# Genome-wide identification of the bZIP transcription factor family and expression analysis under abiotic stress in *Zanthoxylum bungeanum*

**Changming Liu** [1]*, **Zhiguo Tian**[2], **Feng Xian**[3]

**1** Shangluo University, Shangluo, Shaanxi, China, **2** College of Art, Chang Zhou University, Changzhou, China, **3** Inner Mongulia Academy of Agriculture & Animal Husbandry Science, Huhehaote, China

* liuchangming@slxy.edu.cn

## Abstract

*Zanthoxylum bungeanum* (Zb) is an economically and medicinally significant crop that faces numerous environmental stresses due to its broad distribution. Basic leucine zipper (bZIP) transcription factors are extensively involved in plant responses to abiotic stresses and play essential roles in these processes. However, the understanding of bZIP transcription factors in Zb remains limited. In this study, 275 *ZbbZIPs*, which are unevenly distributed across 50 chromosomes and are classified into 13 subfamilies. Each subfamily presents conserved gene structures and motifs. Whole-genome duplication (WGD) and segmental replication events have driven the expansion of *ZbbZIPs*. The ZbbZIP family contains a significant number of elements associated with stress and abscisic acid (ABA) responses, particularly in subfamily A. The codon usage pattern reveals a strong preference for T terminal codons in the ZbbZIP family. Compared with their expression levels under salt stress, the expression levels of the ZbbZIP family were greater under drought and cold stress. Homology annotation and expression profile analyses indicated that *EVM0033673.1* (H, HYH), *EVM0081289.1* (A, DPBF), *EVM0001090.1* (A, DPBF), and *EVM0023876.1* (A, ABF) may significantly contribute to Zb's response to abiotic stresses. These results increase the understanding of the bZIP family and establish a basis for further investigations into the mechanisms by which Zb responds to abiotic stress.

## Introduction

As a small tree in the Rutaceae, *Zanthoxylum bungeanum* (Zb) is widely cultivated as an economic and medicinal plant in China [1]. Its pericarp has a unique numbing taste and can be used as a food seasoning to enhance flavor [2]. Additionally, its leaves, roots, and pericarp are utilized in Chinese medicine to treat diseases [3]. However, its extensive geographical distribution is subject to diverse environmental

**Data availability statement:** All relevant data are within the paper and its Supporting Information files.

**Funding:** This research was funded by Agricultural Innovation and Driven Project of Shaanxi Province, China (No. Shaanxi Agricultural Planning and Finance [2022]29); Shangluo University Industrialization Incubation Project (No. 21CK04)

**Competing interests:** The authors have declared that no competing interests exist.

stresses, especially in the northwestern region of China. The challenges posed by drought, soil salinization, and low winter temperatures threaten the survival of Zb in this region [4,5]. Thus, investigating the adaptive mechanisms of Zb under different abiotic stresses is crucial for the breeding of new varieties.

During development, plants are continuously challenged by abiotic stresses, including drought, salinity, extreme temperatures, and nutrient deprivation [6]. These stressors disrupt critical physiological processes such as photosynthetic efficiency, redox homeostasis, and water and nutrient uptake, ultimately compromising plant development and productivity [7]. To mitigate these adversities, plants have evolved multilevel regulatory networks that integrate molecular, cellular, and biochemical responses [8,9]. Among these mechanisms, transcriptional reprogramming mediated by stress-responsive genes and their regulators, particularly transcription factors (TFs), constitutes a cornerstone of plant stress adaptation [10,11].

The basic leucine zipper (bZIP) family represents a crucial and diverse group of transcription factors in higher plants and is an indispensable part of abiotic stress signal transduction [12]. These transcription factors, characterized by a conserved bZIP domain, may coordinate stress responses by interacting with sugar signaling and various plant hormone pathways [13,14]. The bZIP family plays a crucial role in glucose signal transduction, and mutation or overexpression of the gene products can influence plant sugar synthesis, subsequently impacting the stress response and plant growth [15,16]. For example, the overexpression of *ABF2* within the bZIP domain in *Arabidopsis thaliana* (At) can increase glucose induction and improve the tolerance of plants to abiotic stress [17]. *GmbZIP19* may negatively regulate soybean responses to salt and drought stress through the binding of multiple hormone-induced marker gene promoters [18]. The expression of *VqbZIP39* in At can regulate endogenous ABA synthesis and modulate the expression of multiple stress-induced target genes, thereby increasing tolerance to abiotic stresses [19]. These findings underscore the evolutionary conservation of bZIP-mediated stress adaptation, yet critical knowledge gaps persist in perennial woody species with specialized secondary metabolism.

In this study, we conducted genomic survey of the bZIP gene family in Zb, using transcriptome and genome data, and examined the expression patterns of *ZbbZIPs* under various abiotic stress conditions. Two fundamental questions should be focused in this study: 1) What are the evolutionary differences between the ZbbZIP family and other plants? 2) Do specific *ZbbZIPs* exhibit spatiotemporal expression patterns correlated with drought, salinity, and cold stresses? These results provide valuable insights into the bZIP family and Zb's responses to abiotic stress.

## Materials and methods

### Databases

The *Arabidopsis* Information Resource was used to download AtbZIP protein sequences (http://www.arabidopsis.org/). The Zb genome (BioProject ID: PRJNA524242), as well as transcriptome data for Zb under cold stress (BioProject ID: PRJNA597398), salt stress (BioProject ID: PRJNA1107841), and drought stress

(BioProject ID: PRJNA784034), were downloaded from the National Center for Biotechnology Information (https://www.ncbi.nlm.nih.gov/). The genomes of At and *Citrus clementina* (CCle) were obtained from the Joint Genome Institute https://phytozome-next.jgi.doe.gov/).

### Identification of *ZbbZIPs*

The AtbZIP protein sequences were used as reference sequences. The ZbbZIP protein sequences were initially identified via a genome-wide search of Zb using BLASTp in TBtools software, with a threshold of E < 1e$^{-5}$ [20]. The PfamScan was used to identify the presence of bZIP domains (PF00170) in candidate ZbbZIP sequences (http://pfam.xfam.org/). Additionally, TBtools software was used to calculate the physicochemical properties of the ZbbZIP family members.

### Analysis of sequences alignments and phylogenetic

bZIP protein sequences from At and Zb were aligned using MUSCLE (MEGA11), and a phylogenetic tree was constructed via the maximum-likelihood method (JTT + G4 model; 1,000 bootstrap replicates) [21]. iTOL was used to edit and visualize the phylogenetic tree (https://itol.embl.de/personal_page.cgi).

### Sequence characteristic analysis

MEME Suite 5.4.1 was used to predict the conserved motifs of *ZbbZIPs*. TBtools was utilized to visualize gene structural features and to extract the 2,000 base pairs upstream of the promoter region. PlantCARE was employed to predict the cis-elements (http://bioinformatics.psb.ugent.be/webtools/plantcare/html/).

### Subcellular localization prediction and protein–protein interaction network analysis

The subcellular localization of *ZbbZIPs* was predicted using the Cell-PLoc 2.0 online website (http://www.csbio.sjtu.edu.cn/bioinf/Cell-PLoc-2/). The interactions between *ZbbZIPs* were predicted using the STRING database with the reference species At (https://cn.string-db.org/). Cytoscape was employed to visualize the interactions between *ZbbZIPs*, and betweenness centrality was used to evaluate its importance in the network [22].

### Codon usage pattern and synteny analysis

The CDS of *ZbbZIPs* were analyzed using CodonW 1.4.2 to compute codon preferences based on 14 parameters (http://codonw.sourceforge.net/). The EMBOSS tool was employed to calculate relative synonymous codon usage (RSCU) (https://www.bioinformatics.nl/emboss-explorer). The top 10% of genes with NC values were classified into the low-expression group, and the bottom 10% were classified into the high-expression group. Codons with a ΔRSCU (high-expression group reduced low-expression group) and a total RSCU value greater than 1 were identified as the optimal codons. The SangBox online platform was used to visualize the usage patterns of the optimal codons (http://vip.sanger-box.com/home.html). MCScanX was employed to analyze the syntenic relationships among Zb, At, and CCle [23].

### GO and KEGG enrichment analyses of *ZbbZIPs*

The eggNOG database was utilized for the GO and KEGG annotation of the ZbbZIP family (http://eggnog-mapper.embl.de/). The corresponding enrichment analysis was subsequently conducted using TBtools. The results of the GO and KEGG enrichment analyses were visualized using SRplot (https://www.bioinformatics.com.cn/srplot).

### Plant materials and expression pattern analysis of *ZbbZIPs*

The cultivation and stress treatment of the plant materials were based on the methods of Tian (cold stress), Hu (drought stress) and Nie (salt stress) [2,5,24]. The 'Fuguhuajiao' variety of Zb was selected and cultivated for one year in a

greenhouse maintained at 25 °C, with a humidity of 55–65%. Zb plants were irrigated with 500 mL of 250 mmol/L NaCl solution to induce salinity stress. Leaf samples were harvested at four time points (0, 3, 9, and 24 h). Drought stress was simulated by ceasing irrigatio, and leaves were collected at 0, 3, 9, and 15 d. Additionally, Zb plants were subjected to cold treatment at 4 °C, with leaf samples collected at 0, 1, 3, 6, 12, and 24 h. The collected samples were snap-frozen in liquid nitrogen and stored at −80 °C for qRT–PCR analysis.

TBtools was employed for the processing of raw transcriptome data, which included the generation of FASTQ files, quality control, reference genome alignment, and expression calculation. The expression of *ZbbZIPs* was quantified using the number of fragments per kilobase million (FPKM) values obtained from transcriptome data under drought, cold, and salt stress, and visualized using TBtools.

## qRT-PCR analysis

Primers were designed via Primer Premier 6.0 and validated for specificity using TBtools, ensuring single-band amplification confirmed by melt curve analysis (S1 Table). qRT-PCR experiments were conducted using the method described by Hu [5]. Gene expression levels were standardized to those of *ZbActin* (endogenous control) and quantified through the comparative $2^{-\Delta\Delta Ct}$ method, with triplicate biological samples analyzed for each treatment group [25].

## Statistical analysis

Statistical analyses and standard deviation calculations were conducted using IBM SPSS Statistics 26.

## Results

### Identification and characterization of *ZbbZIPs*

A comprehensive genomic survey of *Zanthoxylum bungeanum* (Zb) identified 275 putative *ZbbZIPs* through conserved domain (Pfam PF00170) and homology-based BLASTp searches against the reference proteome. Chromosomal mapping assigned 268 *ZbbZIPs* to 50 chromosomes, with the remaining 7 genes located on unassembled contigs (Fig 1A and S2 Table). CHR1 shared the highest density of *ZbbZIPs* (25 genes, 9.1% of the total genes). Physicochemical characterization revealed substantial variability among *ZbbZIPs*. The sequence length variation of *ZbbZIPs* ranged from 111 aa (*EVM0063110.1*) to 736 aa (*EVM0093426.1*), representing a 6.6-fold size disparity between the shortest and longest genes. The corresponding molecular weights ranged from 12.11 kDa (*EVM0063110.1*) to 79.48 kDa (*EVM0062482.1*), with the largest protein exceeding the smallest by 6.6-fold. Isoelectric point (pI) analysis demonstrated a predominance of acidic variants, with 63.4% (175/276) of ZbbZIPs exhibiting pI values below 7.0 (S2 Table). Subcellular localization prediction revealed that 92.63% (252) of ZbbZIPs were localized in the nucleus, whereas the remaining 23 genes were distributed among the chloroplast, endoplasmic reticulum, mitochondria, cytoplasm, peroxisomes, and extracellular matrix (S3 Table).

### Gene replication of *ZbbZIPs*

To investigate the expansion mechanism of ZbbZIPs, the potential existence of replication events was analyzed. Tandem duplication events were not identified among the 275 *ZbbZIPs*. The expansion of *ZbbZIPs* was attributed primarily to whole-genome duplication (WGD) and segmental replication, which involved up to 207 *ZbbZIPs*, accounting for 76% of the ZbbZIP family. Additionally, 47 *ZbbZIPs* were influenced by dispersed replication events, representing 17% of all *ZbbZIPs* (Fig 1A and S1 Fig). To understand the expansion process of the ZbbZIP family, the syntenic relationships among CCle, At and Zb were studied. Significant one-to-many relationships were observed between the bZIP members of Zb and those of At and CCle, indicating that the expansion of the ZbbZIP family occurred after the divergence of the entire Rutaceae family (Fig 1B).

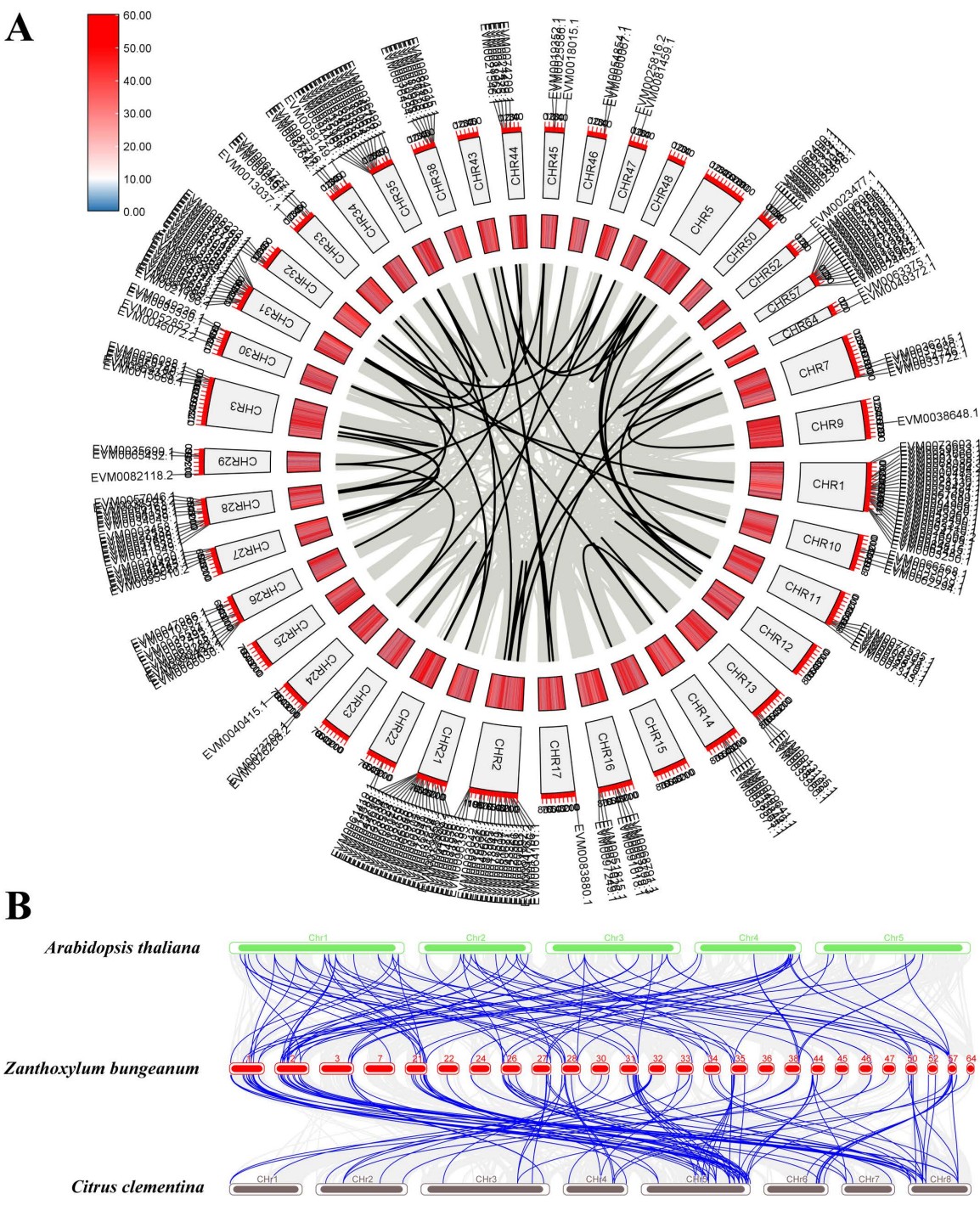

**Fig 1. Synteny analysis of the bZIP gene families. (A)** Collinearity analysis of 275 *ZbbZIPs*. The gray lines link Zb genes that exhibit collinear relationships, and the black lines highlight *ZbbZIPs* associated with fragment duplication events. The outer blocks denote the chromosomes and the inner and central section's depict gene density. **(B)** Collinearity analysis of *bZIPs* between At, CCle, and Zb. The blue lines indicate collinearity between *bZIPs*, while gray lines represent collinearity between species.

## Phylogenetic and structural features of *ZbbZIPs*

The phylogenetic trees were constructed using *AtbZIPs* in conjunction with *ZbbZIPs* to elucidate the evolutionary relationships within the ZbbZIP family. The ZbbZIP family was classified into 13 subfamilies, based on the AtbZIP subfamily. Among them, subfamilies A and D had the greatest number of members, with 56 and 48 members, respectively, which together constituted 37.81% of the entire family. The K subfamily had the lowest number of members with only four (Fig 2).

Comparative analysis of the conserved motifs and exon–intron structures revealed distinct phylogenetic boundaries among the ZbbZIP subfamilies. Motif 1 was generally conserved in 95.3% (262/275) of the families, but was partially absent in subfamily D (13/48). Motif 7 exhibited near-universal retention in non-D subfamilies (223/227, 98% coverage).

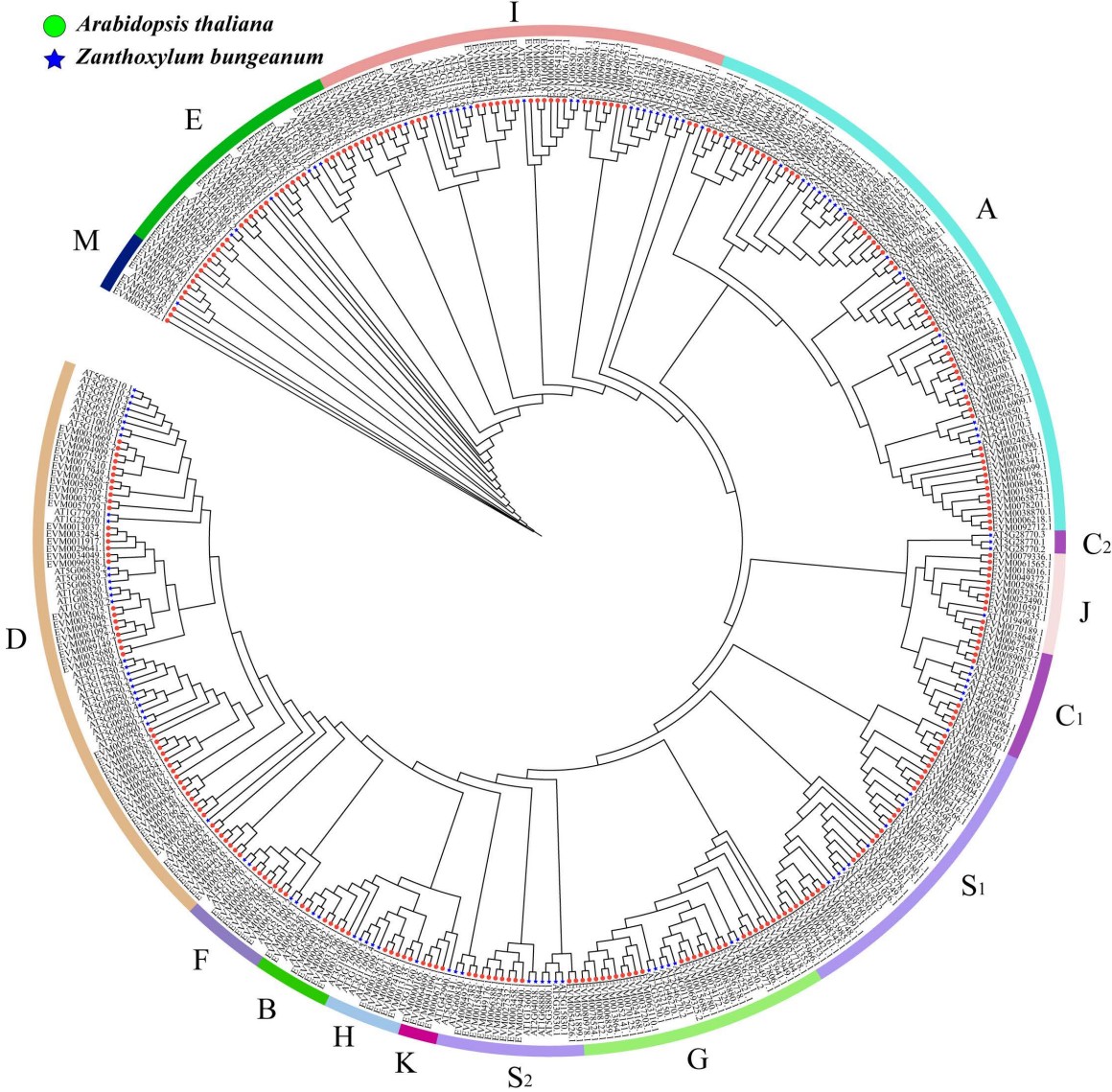

**Fig 2. Phylogeny of the *bZIP* gene family in At and Zb.** The circles indicate the genes from At, and the stars denote the genes from Zb.

Subfamilies A and D exhibited motif structures that were different from those of other subfamilies. Among them, non-A/D subfamily members had only two conserved motifs, motif 1 and motif 7. Subfamily D was distinguished by the absence of motif 7 coupled with the conserved presence of motifs 2–6, while subfamily A contained three conserved motifs 8,9 and 10 (S2A and S3 Figs). Analysis of the gene structure revealed that the number of *ZbbZIP* exons ranged from 2 to 12. Among them, Subfamilies D and G presented the most abundant exons, with averages of 10 and 11, respectively. However, Subfamily S had the lowest abundance of exons, with only one (S2B Fig).

## Promoter regions analysis of *ZbbZIPs*

Twenty-one *cis*-elements of various types were identified within 2000 bp upstream of the CDS of *ZbbZIPs*, excluding promoters, enhancers, and other elements (Fig 3 and S2C Fig). These 21 *cis*-elements were categorized into three groups: growth-related, stress-responsive, and hormone-responsive. The hormone-responsive group comprised five

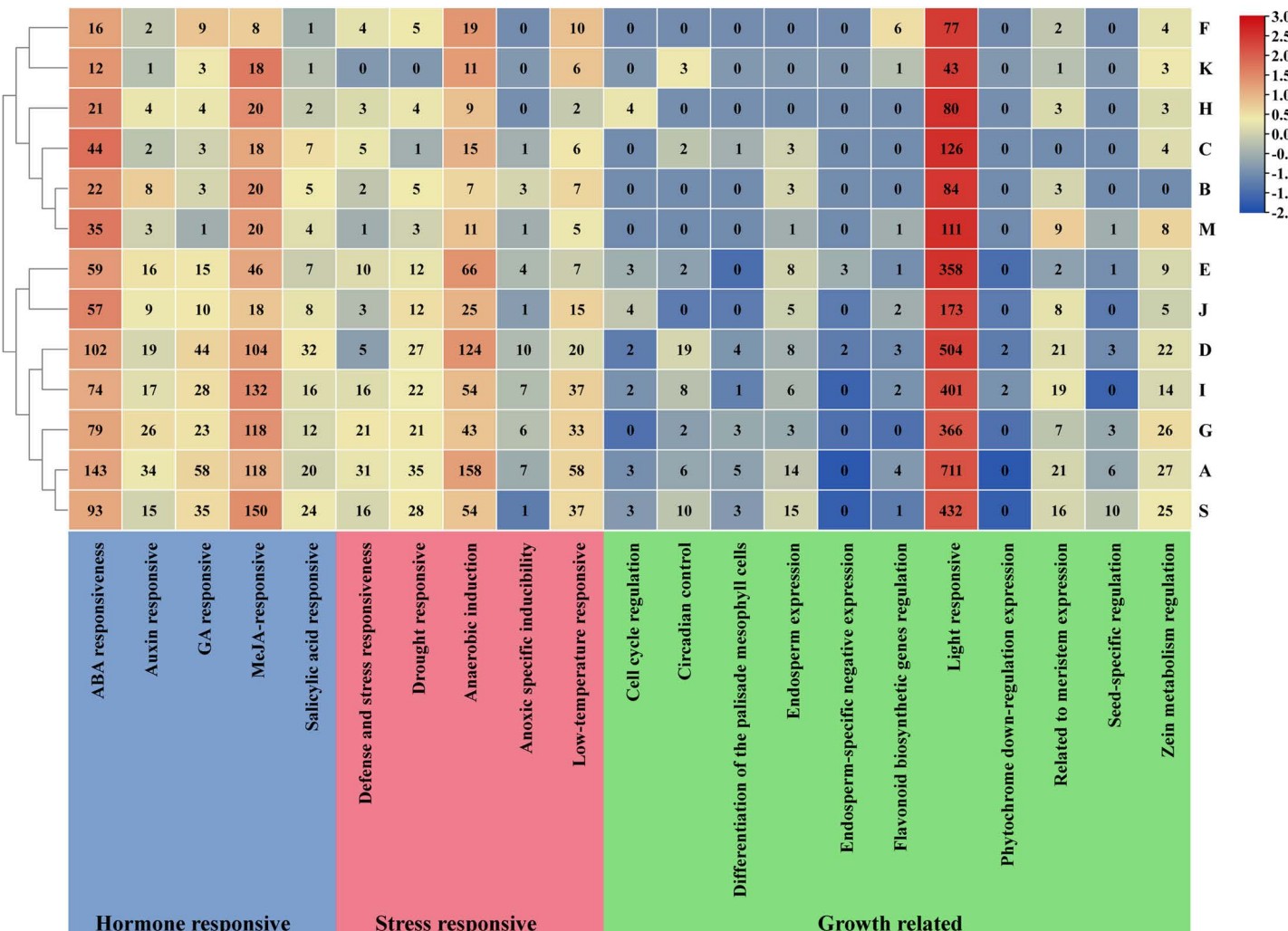

**Fig 3. Number of *cis*-elements in different subfamilies of the ZbbZIP family.** The three colors—blue, pink, and green—represent different types of *cis*-elements. The intensities of the red and blue colors indicate the quantity of cis-elements, with a greater quantity corresponding to a darker shade of red.

*cis*-elements: ABA-responsive, auxin-responsive, salicylic acid-responsive, GA-responsive, and MEJA-responsive. Notably, ABA-responsiveness and MEJA-responsiveness elements were significantly more abundant than the other hormone-responsive elements. The stress-responsive group included five *cis*-elements related to defense and stress responsiveness, drought responsiveness, anaerobic induction, anoxic-specific inducibility, and low-temperature responsiveness. Among these, the number of *cis*-elements associated with anaerobic and low-temperature responsiveness was the greatest in each subfamily. In contrast, the growth-related group contained 11 *cis*-elements, with light-responsive elements being the most prevalent, distributed across nearly every gene.

## Patterns of codon use among *ZbbZPs*

Systematic examination of *ZbbZIPs* codon usage patterns revealed evolutionary constraints shaping their expression potential and functional divergence. The four different base contents in the third base position are shown as follows: T > A > G > C, indicating that members of the ZbbZIP family prefer to end with A/T. The correlation analysis between different codon preference indices revealed that CAI, CBI and Fop were negatively correlated with T3s, but positively correlated with C3s. These results indicate that *ZbbZIPs* have a greater preference for codons ending in T (Fig 4).

The NC map revealed a significant deviation from neutral expectations, with more than 90% of the members lying below the theoretical neutral curve, indicating a strong natural selection influence on the codon (S4A Fig). The PR2-plot analysis revealed that more than 70% of members cluster in regions where A3/(A3 + T3) < 0.5 and where G3/(G3 + C3) < 0.5. These findings suggest a third codon usage preference for *ZbbZIPs* of T > A (S4B Fig). The results from the neutral plot analysis demonstrate that the R2 was considerably less than 1, further indicating that the codons of *ZbbZIPs* are influenced by natural selection, leading to a composition of the third base that differed from those of the first and second bases (S4C Fig). Overall, these results suggest that codon evolution in *ZbbZIPs* is more strongly shaped by natural selection, with a pronounced preference for T-ending codons.

The analysis of codon usage revealed that the number of optimal codons varied among the different subfamilies of Zbb-ZIP, ranging from 10 to 22. Among these families, the A and S families presented the highest optimal codon counts, with 20 and 22 optimal codons, respectively. Additionally, two-thirds of the tripartite high-frequency codons (GGT, AAT, AGG) presented a conserved T-terminal architecture, which was consistent with the established T3 bias in this gene family (S5 Fig).

## GO and KEGG enrichment analyses of *ZbbZIPs*

The functional annotation of ZbbZIP family members through Gene Ontology (GO) and KEGG pathway analyses revealed potential biological functions. GO enrichment analysis revealed a significant enrichment of pathways associated with ABA signaling (GO:0009737) and the cellular response to stimuli (GO:0071214) (S6A Fig). Additionally, KEGG analysis revealed that environmental information processing (ko02010) and plant hormone signal transduction pathways (ko04075) were enriched in the ZbbZIP family. Subfamilies I and H exhibited specific enrichment in the circadian rhythm pathway (ko04712) and the MAPK signaling pathway (S6B Fig).

## Expression profile of *ZbbZIPs*

Transcriptomic profiling of the ZbbZIP family under drought, salt, and cold stresses revealed distinct temporal expression dynamics. Genes with FPKM > 10 were defined as the high marker genes for analysis in this study. High-expression *ZbbZIPs* (FPKM >10 threshold) were stratified into four distinct expression groups (I–IV) under drought stress, which presented inverse expression patterns between groups I (upregulated) and II (downregulated). The peak expression level in groups III and IV were concentrated at 6–9 d (Fig 5A). Cold stress similarly generated four expression groups, with groups II and III showing expression peaks during the late-phase (12–24 d) and early-phase (0–1 d) stress treatments, respectively (Fig 5B). However, the expression of *ZbbZIPs* was more strongly inhibited by salt stress, and significantly fewer genes were highly expressed under salt stress than under drought and cold stress. The expression trends of these

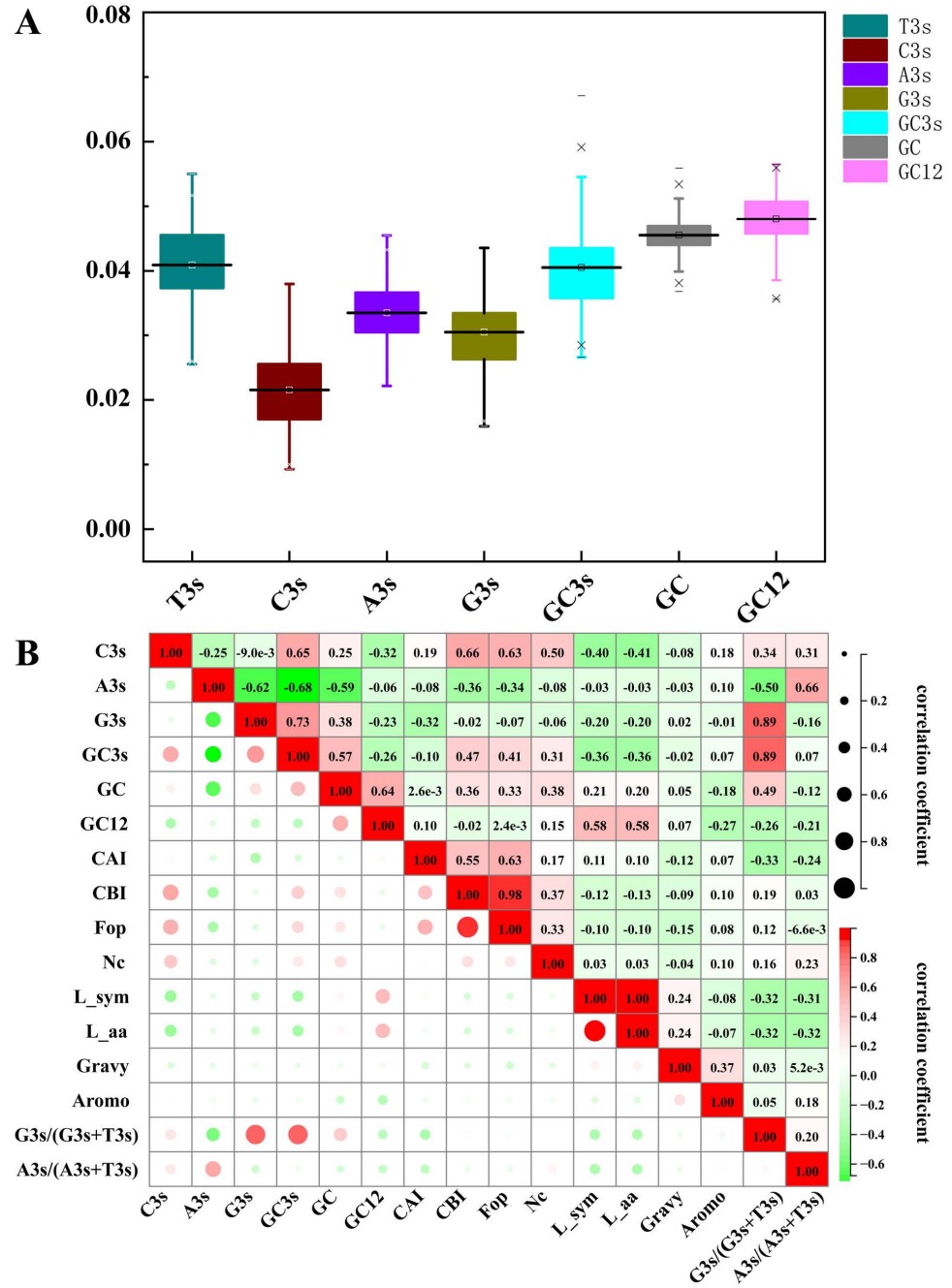

**Fig 4. Base composition of the ZbbZIP gene family. (A)** Schematic representation of the nucleotide composition in *ZbbZIPs*. **(B)** Pearson correlation analysis of codon usage bias parameters in *ZbbZIPs*.

genes could be divided into two groups, in which group II genes were upregulated and peaked after 3–9 h of salt stress treatment. (Fig 5C).

EVM0023876.1 and EVM0033673.1 presented high expression levels under the three stress conditions. Furthermore, 14 other members presented high expression levels in response to drought and cold stresses (Fig 6A). Protein–protein

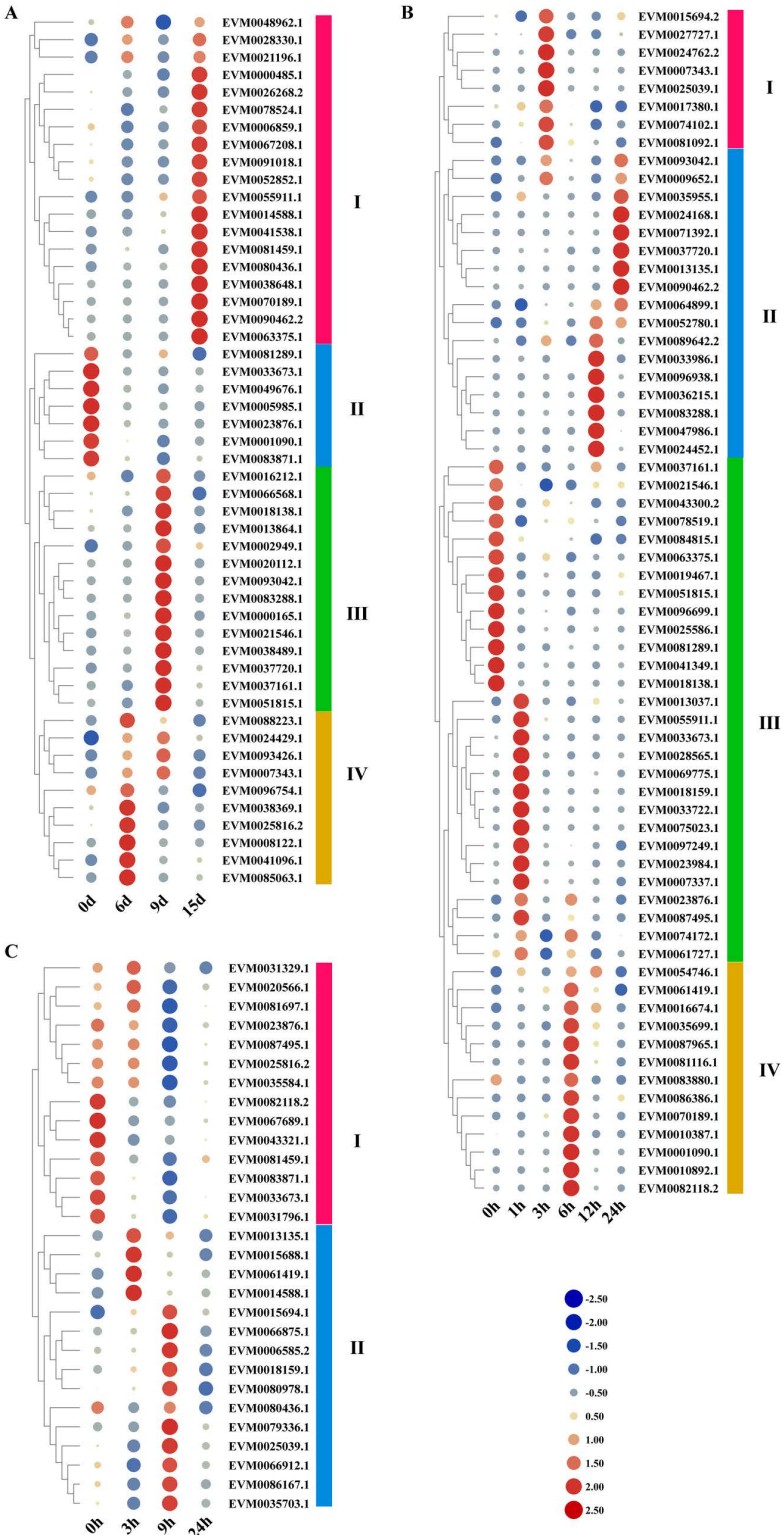

**Fig 5. Expression profiles of the ZbbZIP gene family under abiotic stress. (A)** Expression profiles of *ZbbZIPs* under drought stress. **(B)** Expression profiles of *ZbbZIPs* under cold stress. **(C)** Expression profiles of *ZbbZIPs* under salt stress. The varied color bars on the right differentiate between distinct groups. The scale reflects the relative intensity of FPKM values, with a gradient from blue to red signifying an increase in expression levels.

interaction analysis of the ZbbZIP family revealed numerous and complex regulatory networks among *ZbbZIPs*. *EVM0033673.1* was highly expressed across multiple stress conditions and exhibited a mutual regulatory relationship with other family members. Additionally, *EVM0081289.1*, *EVM0083288.1*, and *EVM0063375.1* responded to both drought and cold stresses, suggesting potential interactions among them (Fig 6B). To gain a deeper understanding of the potential functions of these highly expressed genes, their homologous genes in At were annotated. Twelve genes were found to have homologous counterparts with well-defined functions, whereas four genes (*EVM0001090.1*, *EVM0023876.1*, *EVM0033673.1*, *EVM0081289.1*) were associated with abscisic acid (ABA) signaling and stress response (S4 Table). Based on the results of the stress response analysis, interaction network, and homologous gene annotation, *EVM0033673.1* (H, HYH), *EVM0081289.1* (A, DPBF), *EVM0001090.1* (A, DPBF), and *EVM0023876.1* (A, ABF) may play significant roles in the responses of Zb to various stressors.

qRT–PCR analysis was employed to further examine the responses of the core genes to abiotic stress. Both expression profiles demonstrated downregulated expression of core genes subjected to drought stress. The expression of the core genes determined by qPCR increased at 9 h after drought treatment. After cold stress treatment, the expression peaks of the two expression profiles, especially those of *EVM0033673.1* and *EVM0001090.1*, appeared at different treatment times. However, the trends of the two expression profiles under salt stress were generally consistent. Both expression profiles of *EVM0081289.1* were low at 9 h after stress treatment. These results indicated that there were differences in the overall expression trends between the two methods. This error may have been caused by the plant materials and experimental methods used (Fig 7).

## Discussion

As one of the largest and most functionally diverse transcription factor families in plants, the bZIP transcription factor family has been widely reported across numerous species [26]. A comprehensive analysis revealed 275 *bZIP* members in the Zb genome, markedly exceeding the numbers observed in *Oryza sativa* (89 members, monocots), At (78 members, dicots), *Populus trichocarpa* (86 members, woody plants), *Citrus sinensis* (50 members, Rutaceae family), and *Gossypium hirsutum* (205 members, tetraploid) [27–30]. Zb, an allotetraploid woody plant, has a significantly larger genome and gene

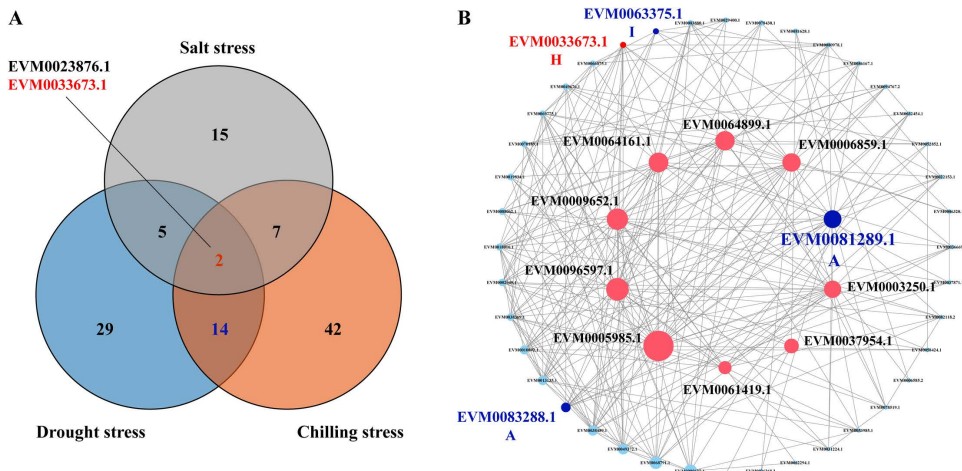

**Fig 6. Selection of core genes encoding *ZbbZIPs* in response to abiotic stress. (A)** Venn diagram of highly expressed genes under three abiotic stresses; **(B)** Protein-protein interaction network between *ZbbZIPs*. Pink genes indicate genes with high centrality in the network mediation number. Blue numbers and genes indicate genes highly expressed under drought and cold stress. Red genes indicate *ZbbZIPs* that are highly expressed under all three stresses.

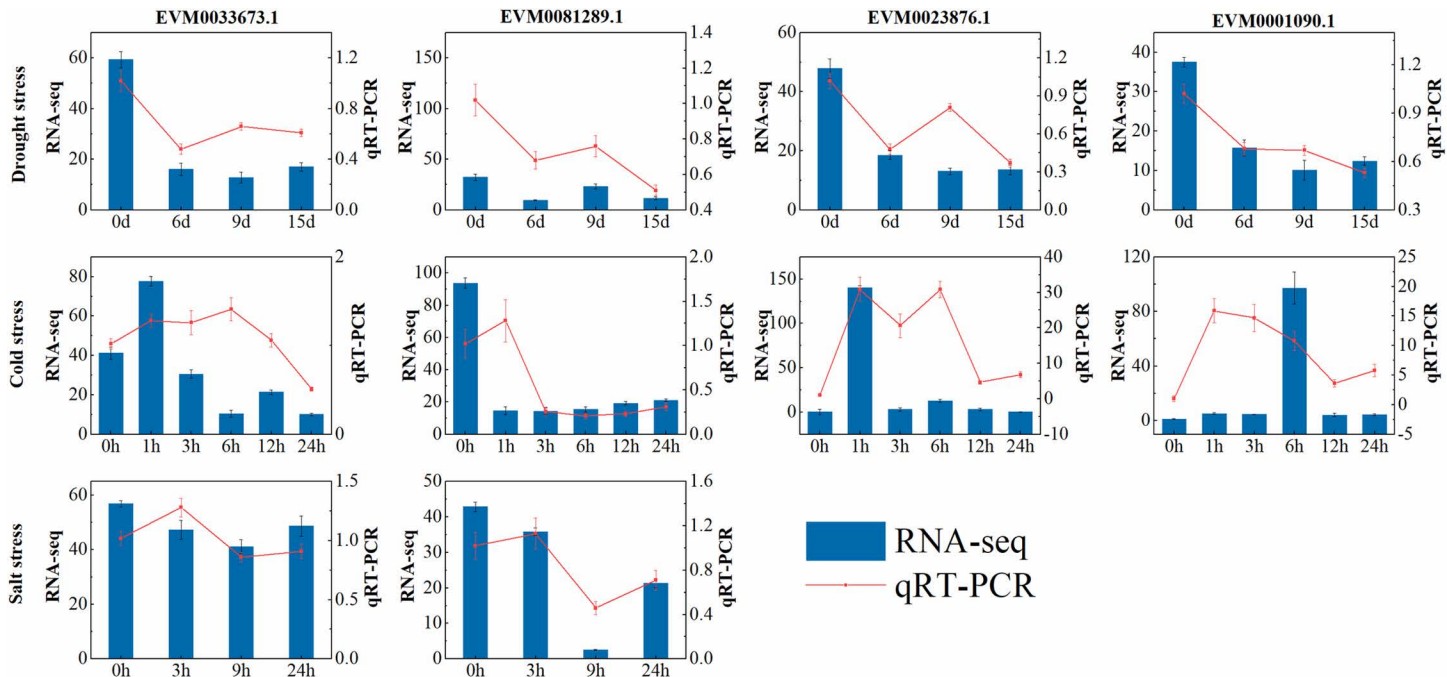

**Fig 7. Expression profiles of core genes associated with abiotic stress.** The blue bar represents the RNA-seq expression levels based on the FPKM values, and the red line indicates the relative expression levels determined by qRT-PCR analysis.

capacity than do common diploid plants [15]. The results of this study indicated that the ZbbZIP family has undergone significant expansion over the course of evolution. For polyploid plants, genome doubling or WGD, as well as segmental duplication, are the primary mechanisms driving gene family expansion [31]. These findings were also corroborated by this study of duplication events, which revealed that WGD and fragment duplications accounted for 76% of ZbbZIP family expansions. Notably, no tandem duplication events were detected in the entire ZbbZIP family. A previous analysis of multiple Zb gene families has shown that tandem duplication events affected the expansion of the family [32]. Tandem duplication is often one of the key mechanisms driving the increase in gene number and functional diversification, which has a positive effect on adaptation to the environment [33]. The lack of the driving force caused by tandem repeats in the ZbbZIP family may make it more conserved in function and evolution.

In this study, 275 *ZbbZIP* members were classified into 13 subfamilies, which is consistent with the classification in At [34]. These findings indicated that Zb has retained all subfamilies throughout its evolutionary history. Compared with the AtbZIP family, the ZbbZIP family has expanded by more than threefold. However, this expansion was not uniformly distributed across all subfamilies; notably, subfamilies M and J exhibited a greater degree of expansion than their single-gene counterparts in At. These genes may play a more significant role in enabling Zb to adapt to external environmental conditions.

The gene structure and motifs indicate conservation within the subfamily as well as evolutionary differences among the subfamilies in Zb. Notably, the A and D subfamilies, two major groups within the ZbbZIP gene family, each possess unique motif structures that are absent in other subfamilies. These distinctive motifs may play a crucial role in their specific functions related to environmental adaptation. Within the bZIP family of At, the A subfamily consists primarily of DC3 promoter binding factors (DPBF) and ABCB1 response element binding factor (ABFs). Both factors play crucial roles in abscisic acid (ABA) signaling and in regulating plant responses to various abiotic stresses [35,36]. Conversely, subfamily D genes are classified as TGAs, which are involved in the plant immune response as part of systemic acquired resistance

[37]. Furthermore, the number of introns and exons in subfamily S were significantly lower than those in the other subfamilies. Introns are typically found in ancient eukaryotes, and as organisms evolve, the number of introns tends to decrease. Consequently, it is often posited that ancient sequences may contain greater numbers of introns [38]. The S subfamily is likely the younger subfamily within the entire ZbbZIP family.

The type and quantity of cis-elements directly influence gene transcription, spatiotemporal specific expression, and biological adaptability to environmental changes [39]. Notably, light-responsive cis-elements are universally present in the promoter regions of all *ZbbZIPs*. As both the primary energy source and a critical signaling regulator in nature, light profoundly influences plant growth and developmental processes [40]. This regulatory role aligns with the inherent circadian rhythms of plants, where light-mediated control of gene expression coordinates physiological and metabolic adaptations [41,42]. Intriguingly, our findings specifically revealed that identified genes from the H subgroup were enriched in circadian rhythm pathways. This observation is consistent with the established function of H subfamily members (*AtHY5* and *AtHYH*) in photomorphogenesis and photosynthetic pigment biosynthesis [34]. In addition, subfamily A contained the greatest number of the ABA-responsiveness, drought responsive, and low temperature cis-elements. The ABA signal transduction pathway is a crucial means by which plants respond to environmental stresses, especially abiotic stresses [43]. ABF and DPBF (subfamily A) are often regulated and phosphorylated by ABA and various stresses, and then binds to the ABRE of downstream genes (drought responsiveness) to induce their expression to regulate responses to various stresses in plant [44].

The differential expression patterns of plant gene families in under various stress conditions reveal the dynamic adaptation mechanisms to environmental challenges [45]. Under drought and cold stress, *ZbbZIPs* presented elevated expression levels, whereas high salt stress resulted in the downregulation of *ZbbZIPs*. These findings suggest that Zb prioritizes drought/cold adaptation over salt tolerance—a hypothesis corroborated by parallel suppression of the ZbbHLH and ZbSPL families under salt stress [46,47]. In this study, we identified four core abiotic stress-responsive genes in Zb: three A-subfamily members (*EVM0081289.1*/DPBF, *EVM0023876.1*/ABF, and *EVM0001090.1*/DPBF) and one H-subfamily member (*EVM0033673.1*). The A-subfamily homologs play conserved roles in ABA signaling, as demonstrated by *HvABI5*'s regulation of drought-responsive *HVA1*/*HVA22* in barley [48] and AtABF3's involvement in JA signaling through the *JAZ1-MYC2* interaction in At [49]. In addition, the H-subfamily *EVM0033673.1*, which is homologous to *AtHYH*, modulates abiotic stress responses via temperature-sensitive *MIR169* regulation and ABA pathway coordination [34,49]. In conclusion, the regulation of *ZbbZIPs* in response to abiotic stress in Zb may be based on ABA signal transduction.

## Conclusions

A total of 275 *ZbbZIPs* were identified within the Zb genome and were classified into 13 subfamilies. Conserved gene structures and motifs were observed within these subfamilies. The expansion of the ZbbZIP family was influenced by whole-genome replication (WGD) and segmental replication. Codons ending in T were favored during evolution within the ZbbZIP family. Promoter analysis revealed that *cis*-elements responsive to stress and ABA hormones were enriched in the promoter regions, particularly in subgroup A. Expression profiling under various abiotic stresses demonstrated that the ZbbZIP family presented relatively high expression levels in response to drought and cold stress. *EVM0033673.1* (H, HYH), *EVM0081289.1* (A, DPBF), *EVM0001090.1* (A, DPBF), and *EVM0023876.1* (A, ABF) may play crucial roles in the response of Zb to drought and cold stress. These results provide a foundation for further understanding the mechanisms by which the ZbbZIP family responds to abiotic stress.

## Supporting information

**S1 Fig. Gene replication events of the bZIP gene family in *Zanthoxylum bungeanum*.** The same color in the pie chart and the histogram represents the same replication event.
(TIFF)

**S2 Fig. Gene sequence characteristics of the bZIP gene family in *Zanthoxylum bungeanum*.** (A) Conserved motif of *ZbbZIPs*. (B) Gene structure of *ZbbZIPs*. (C) *Cis*-elements of *ZbbZIPs*. The letters after the gene indicate the subfamily from which it comes.
(TIFF)

**S3 Fig. Structural information of the conserved motifs in *ZbbZIPs*.**
(TIFF)

**S4 Fig Analysis of the codon preference of the bZIP gene family in *Zanthoxylum bungeanum*.** (A) ENC map analysis. (B) PR2 plot analysis. (C) Neutral plot analysis.
(TIFF)

**S5 Fig. Optimal codon analysis of the bZIP gene family in *Zanthoxylum bungeanum*. The bar chart (top) represents the number of italicized codons below.** The black dots in the graph (bottom) indicate that the corresponding codon is the optimal codon of the corresponding subfamily. The measured color bands and numbers on the left represent the number of optimal codons in the corresponding subfamily.
(TIFF)

**S6 Fig. GO and KEGG enrichment analyses of *ZbbZIPs*.** (A) GO enrichment analysis of *ZbbZIPs*. (B) KEGG enrichment analysis of *ZbbZIPs*.
(TIFF)

**S1 Table. Primers of core genes for qRT–PCR.**
(XLSX)

**S2 Table. Classification and characterization of the putative the bZIP gene family in *Zanthoxylum bungeanum*.**
(XLSX)

**S3 Table. Subcellular localization (except for the nucleus) of the bZIP gene family in *Zanthoxylum bungeanum*.**
(XLSX)

**S4 Table. Orthologous gene annotation results of high-level *ZbbZIPs* under multiple stress conditions.**
(XLSX)

## Author contributions

**Data curation:** Zhiguo Tian.

**Funding acquisition:** Changming Liu.

**Investigation:** Feng Xian.

**Methodology:** Zhiguo Tian, Feng Xian.

**Software:** Feng Xian.

**Writing – original draft:** Changming Liu.

**Writing – review & editing:** Changming Liu.

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
