## [Decision Letter · Decision Letter 0]

8 Apr 2025

PONE-D-25-14248Genome-Wide Identification of bZIP Transcription Factors Family and Analysis of Its Expression Under Abiotic Stress in Zanthoxylum BungeanumPLOS ONE

Dear Dr. Liu,

Thank you for submitting your manuscript to PLOS ONE. After careful consideration, we feel that it has merit but does not fully meet PLOS ONE’s publication criteria as it currently stands. Therefore, we invite you to submit a revised version of the manuscript that addresses the points raised during the review process.

We look forward to receiving your revised manuscript.

Kind regards,

Shailender Kumar Verma, Ph.D.

Academic Editor

PLOS ONE

 [This research was funded by Agricultural Innovation and Driven Project of Shaanxi Province, China (No. Shaanxi Agricultural Planning and Finance [2022]29); Shangluo University Industrialization Incubation Project (No. 21CK04)]. 

4. Please amend your authorship list in your manuscript file to include author Feng Xian ,.

Additional Editor Comments (if provided):

Reviewers' comments:

Reviewer's Responses to Questions

**Comments to the Author**

1. Is the manuscript technically sound, and do the data support the conclusions?

Reviewer #1: Yes

Reviewer #2: Partly

2. Has the statistical analysis been performed appropriately and rigorously? 

Reviewer #1: Yes

Reviewer #2: No

3. Have the authors made all data underlying the findings in their manuscript fully available?

Reviewer #1: Yes

Reviewer #2: Yes

4. Is the manuscript presented in an intelligible fashion and written in standard English?

Reviewer #1: Yes

Reviewer #2: Yes

5. Review Comments to the Author

Reviewer #1: This manuscript focused on bioinofrmatic analyses of bZIP transcription factors in zanthoxylum bungeanum, and provided a comprehensive acquaintance of molecular mechanism of bZIP genes in response to multiple abiotic stresses. Generally, we believe this manuscript are of innovation and significance for bZIP studies, while many technical issues must be put forward first.

(1)The writing of the manuscript needs improvement, and suggest to invite one professional researcher to polish this article.

(2)Line 163, revise 21 as twenty-one, since the sentences should start with capital letters.

(3)We noticed that the authors performed qRT-PCR experiments on the chosen bZIP genes for expression-pattern investigation, however, the consistency between qRT-PCR and RNA-seq was low. Suggest to re-chose some ones to conduct qRT-PCR experiment, or draw the correlation coefficient figure.

(4)Why not to perform VIGS experiment to verify the potential genes ?

Reviewer #2: The manuscript mainly identifies the bZIP gene family, but due to the large number of members, the authors did not do some common bioinformatics analyses such as gene structure, physicochemical properties and chromosomal localisation, but the current authors' collation of some databases also provides data support for the identification of the bZIP gene family of Zanthoxylum Bungeanum. The details are as follows:

1. The author was advised to change the title; for example: “Genome-wide identification of the bZIP transcription factor family and expression analysis under abiotic stress in Zanthoxylum Bungeanum.”

2. Lines 19-20: Please replace ‘higher activity’ with ‘high expression’ or ‘higher transcript abundance’. Please check the full text for similar issues.

3. Please add ‘bioinformatics analyses’ to keywords section.

4. The authors were advised to change the order of the introduction section. The authors were asked to first describe the abiotic stresses encountered in growing peppers (third paragraph of the introduction), then go on to describe the plant damage of the abiotic stresses (first paragraph of the introduction), then describe the role of bZIP in the abiotic stresses (second paragraph of the introduction), and finally lead in with what was done in the study? What was the significance of doing the study? Please ask the authors to be aware of the connection and logic between each paragraph in their descriptions.

5. The methods of data analysis are missing from the Materials and Methods section. Authors are requested to check for formatting problems in the subheadings.

6. It is recommended that the authors explain in the materials and methods why 250 mmol/L NaCl solution was chosen and why different time points were chosen to collect the samples.

7. It is suggested that the author modify the representation of the figure notes in the result part, some of which are abbreviated, and some are full names. It is suggested that the author carefully modify according to the format requirements of the journal

8. The authors were advised to re-describe the results section; the current description is too heavily colloquial and unlike the form of writing a scientific paper. For example: Lines 245-248. The authors can specifically describe how many-fold the expression of bZIP genes is repressed under abiotic stress and how they trend. Please ask the author to check the tenses throughout the text.

6. PLOS authors have the option to publish the peer review history of their article (what does this mean? ). If published, this will include your full peer review and any attached files.

**Do you want your identity to be public for this peer review?** For information about this choice, including consent withdrawal, please see our Privacy Policy .

Reviewer #1: No

Reviewer #2: No

---

## [Author Response · Author response to Decision Letter 1]

21 Apr 2025

Manuscript ID: PONE-D-25-14248

MS TITLE: Genome-Wide Identification of bZIP Transcription Factors Family and Analysis of Its Expression Under Abiotic Stress in Zanthoxylum Bungeanum

Dear editor and reviewers,

We sincerely appreciate the critical comments and thoughtful suggestions provided by you and the reviewers, as they have greatly contributed to revising and improving our paper, as well as guiding our future research endeavors. We have thoroughly examined the comments and made necessary corrections in accordance with your feedback. Please refer to my itemized responses below and find the revised/corrected versions in the re-submitted files.

Editor:

Question 1. Please ensure that your manuscript meets PLOS ONE's style requirements, including those for file naming.

Answer: According to your suggestion, we have made corrections according to the style requirements of the journal, adjusted the font size of the title, and removed the number prefix of the title, for example: 1. Introduction was changed to Introduction. And the title is capitalized with the first letter of the word. Change the icon format in the support information, for example Fig S1 to S1 Fig. If there is any need for more modification, we will further improve it according to the subsequent suggestions.

Question 2. Please state what role the funders took in the study. If the funders had no role, please state: "The funders had no role in study design, data collection and analysis, decision to publish, or preparation of the manuscript." If this statement is not correct you must amend it as needed.

Answer: The two funders are the source of funding for the study. They are not specifically involved in research design, data collection and analysis, publication decisions, or manuscript preparation. We have disclosed the funding statement in our cover letter and manuscript. Please check these contents.

Question 3. PLOS requires an ORCID iD for the corresponding author in Editorial Manager on papers submitted after December 6th, 2016. Please ensure that you have an ORCID iD and that it is validated in Editorial Manager.

Answer: Thank you for reminding us that we have associated the ORCID ID before submitting the revised manuscript.

Question 4. Please amend your authorship list in your manuscript file to include author Feng Xian.

Answer: We are sorry that due to an oversight and mistake, one of authors was not mentioned in the manuscript. We have included the author in the manuscript (Line 3 in revised manuscript).

Reviewer #1: This manuscript focused on bioinofrmatic analyses of bZIP transcription factors in zanthoxylum bungeanum, and provided a comprehensive acquaintance of molecular mechanism of bZIP genes in response to multiple abiotic stresses. Generally, we believe this manuscript are of innovation and significance for bZIP studies, while many technical issues must be put forward first.

Question 1. The writing of the manuscript needs improvement, and suggest to invite one professional researcher to polish this article.

Answer: Thank you for your thoughtful advice. We have engaged two professional researchers to optimize the writing of our manuscript. Proof of retouching from this institution is on the last page of this document.

Question 2. Line 163, revise 21 as twenty-one, since the sentences should start with capital letters.

Answer: Your suggestion is reasonable. We have modified 21 to twenty-one (Line 177 in revised manuscript).

Question 3. We noticed that the authors performed qRT-PCR experiments on the chosen bZIP genes for expression-pattern investigation, however, the consistency between qRT-PCR and RNA-seq was low. Suggest to re-chose some ones to conduct qRT-PCR experiment, or draw the correlation coefficient figure.

Answer: Thank you for the suggestion. Our core genes showed differences in expression trends between qRT-PCR and RNA-seq. Although we used the same cultivars, breeding methods, and stress treatments, there may be differences in genetic background from those used for RNA-seq. In particular, the transcriptomes involved are from three different experiments. This difference in genetic background may have contributed to this error. Secondly, there are also differences between the two technical measures, which will affect the consistency of their expression trends. However, it is worth affirming that the overall trend of our qRT-PCR is consistent with that of RNA-seq.

Question 4. Why not to perform VIGS experiment to verify the potential genes?

Answer: Thank you for your constructive suggestions. In fact, Zanthoxylum Bungeanum is an allotetraploid plant, and there is a lack of mature technology for VIGS experiments. At present, we lack the experience of gene verification work, and we will gradually try to verify its possibilities on model plants in the future.

Reviewer #2: The manuscript mainly identifies the bZIP gene family, but due to the large number of members, the authors did not do some common bioinformatics analyses such as gene structure, physicochemical properties and chromosomal localisation, but the current authors' collation of some databases also provides data support for the identification of the bZIP gene family of Zanthoxylum Bungeanum.

Question 1. The author was advised to change the title; for example: “Genome-wide identification of the bZIP transcription factor family and expression analysis under abiotic stress in Zanthoxylum Bungeanum.

Answer: Thank you for your valuable advice. We have changed the title to "Genome-wide identification of the bZIP transcription factor family and expression analysis under abiotic stress. in Zanthoxylum Bungeanum". (Lines 1-2 in revised manuscript)

Question 2. Lines 19-20: Please replace ‘higher activity’ with ‘high expression’ or ‘higher transcript abundance’. Please check the full text for similar issues.

Answer: According to your suggestion and the polishing suggestions of professional researchers, we will modify the corresponding content. We hope our modification conforms to your ideas, please refer to Lines 19-20 for details.

Question 3. Please add ‘bioinformatics analyses’ to keywords section.

Answer: Thank you for your thoughtful advice. We have added bioinformatics analysis to the keyword section (Line 25 in revised manuscript).

Question 4. The authors were advised to change the order of the introduction section. The authors were asked to first describe the abiotic stresses encountered in growing peppers (third paragraph of the introduction), then go on to describe the plant damage of the abiotic stresses (first paragraph of the introduction), then describe the role of bZIP in the abiotic stresses (second paragraph of the introduction), and finally lead in with what was done in the study? What was the significance of doing the study? Please ask the authors to be aware of the connection and logic between each paragraph in their descriptions.

Answer: Thank you for giving us these good suggestions. As requested, we have adjusted the order of the paragraphs in introduction section. In addition, the reference numbers have also been adjusted accordingly (Lines 27-53 in revised manuscript). We hope that the revised paragraph structure will meet with your approval.

Question 5. The methods of data analysis are missing from the Materials and Methods section. Authors are requested to check for formatting problems in the subheadings.

Answer: According to your suggestion, here we refer to other relevant studies [1-2], optimize this part, and supplement the relevant content (Lines 70-71, 92-95, 104-105, and 122-123 in revised manuscript). We hope that the corresponding changes to your question can be approved by you and if this part needs further improvement, we would like you to point out the specific problems. Thank you very much.

[1] Wang S, Hu W, Zhang X, Liu Y, Liu F. Identification and Characterization of SQUAMOSA Promoter Binding Protein-like Transcription Factor Family Members in Zanthoxylum bungeanum and Their Expression Profiles in Response to Abiotic Stresses. Plants [Internet]. 2025; 14(4).

[2] Du H, Feng BR, Yang SS, Huang YB, Tang YX. The R2R3-MYB Transcription Factor Gene Family in Maize. Plos One. 2012;7(6). doi: 10.1371/journal.pone.0037463. PubMed PMID: WOS:000305351700006.

Question 6. It is recommended that the authors explain in the materials and methods why 250 mmol/L NaCl solution was chosen and why different time points were chosen to collect the samples.

Answer: Thank you for the suggestion, and we cited the sources of three stress treatment methods in the section of “Plant materials and expression pattern analysis of ZbbZIPs” (Lines 104-105 in revised manuscript: The cultivation and stress treatment of the plant materials were based on the methods of Tian (cold stress), Hu (drought stress) and Nie (salt stress) [2,5,24]).

Question 7. It is suggested that the author modify the representation of the figure notes in the result part, some of which are abbreviated, and some are full names. It is suggested that the author carefully modify according to the format requirements of the journal

Answer: Thank you for your thoughtful suggestion. We modified the figure notes in the result part and unified them into "Fig" according to the requirements of the journal (Lines 222, 225 and 274 in revised manuscript).

Question 8. The authors were advised to re-describe the results section; the current description is too heavily colloquial and unlike the form of writing a scientific paper. For example: Lines 245-248. The authors can specifically describe how many-fold the expression of bZIP genes is repressed under abiotic stress and how they trend. Please ask the author to check the tenses throughout the text.

Answer: According to your suggestion, we have redescribed the results section, and at the same time invited two professional researchers to optimize the writing of our manuscript. Proof of retouching from this institution is on the last page of this document. In particular, the sections you mentioned (Lines 245-248) are scientifically detailed (Lines 264-272 in revised manuscript). We hope that the revision of this manuscript can receive your approval.

Once again, thank you very much for your kind comments and suggestions.

We look forward to your information about our revised paper.

Best wishes for you!

Yours sincerely,

Changmin Liu

---

## [Decision Letter · Decision Letter 1]

25 Apr 2025

Genome-wide identification of the bZIP transcription factor family and expression analysis under abiotic stress in Zanthoxylum bungeanum

PONE-D-25-14248R1

Dear Dr. Liu,

We’re pleased to inform you that your manuscript has been judged scientifically suitable for publication and will be formally accepted for publication once it meets all outstanding technical requirements.

Kind regards,

Shailender Kumar Verma, Ph.D.

Academic Editor

PLOS ONE

Additional Editor Comments (optional):

Reviewers' comments:

Reviewer's Responses to Questions

**Comments to the Author**

1. If the authors have adequately addressed your comments raised in a previous round of review and you feel that this manuscript is now acceptable for publication, you may indicate that here to bypass the “Comments to the Author” section, enter your conflict of interest statement in the “Confidential to Editor” section, and submit your "Accept" recommendation.

Reviewer #1: All comments have been addressed

Reviewer #2: All comments have been addressed

2. Is the manuscript technically sound, and do the data support the conclusions?

Reviewer #1: Yes

Reviewer #2: Yes

3. Has the statistical analysis been performed appropriately and rigorously? 

Reviewer #1: Yes

Reviewer #2: Yes

4. Have the authors made all data underlying the findings in their manuscript fully available?

Reviewer #1: Yes

Reviewer #2: Yes

5. Is the manuscript presented in an intelligible fashion and written in standard English?

Reviewer #1: Yes

Reviewer #2: Yes

6. Review Comments to the Author

Reviewer #1: This manuscript focused on bioinofrmatic analyses of bZIP transcription factors in zanthoxylum bungeanum, and provided a comprehensive acquaintance of molecular mechanism of bZIP genes in response to multiple abiotic stresses. Generally, we believe this manuscript are of innovation and significance for bZIP studies, and all the incorrect portions have been modified in this manuscript. The revised article has met the standard for publication.

Reviewer #2: (No Response)

7. PLOS authors have the option to publish the peer review history of their article (what does this mean? ). If published, this will include your full peer review and any attached files.

**Do you want your identity to be public for this peer review?** For information about this choice, including consent withdrawal, please see our Privacy Policy .

Reviewer #1: No

Reviewer #2: No

---

## [Editor Report · Acceptance letter]

PONE-D-25-14248R1

PLOS ONE

Dear Dr. Liu,

I'm pleased to inform you that your manuscript has been deemed suitable for publication in PLOS ONE. Congratulations! Your manuscript is now being handed over to our production team.

Kind regards,

on behalf of

Dr. Shailender Kumar Verma

Academic Editor

PLOS ONE